# A prospective, multicenter, post-marketing observational study to measure the quality of life of HCV genotype 1 infected, treatment naïve patients suffering from fatigue and receiving 3D regimen: The HEMATITE study

Nasser Semmo[1], Beat Müllhaupt[2], Lisa Ruckstuhl[3], Lorenzo Magenta[4], Olivier Clerc[5], Ralph Torgler[3], David Semela[6]*

1 Hepatology, Department of BioMedical Research, University of Bern, Bern, Switzerland, 2 Division of Gastroenterology and Hepatology, University Hospital Zürich, Zürich, Switzerland, 3 AbbVie Schweiz AG, Baar, Switzerland, 4 Fondazione Epatocentro Ticino, Lugano, Switzerland, 5 Infectious Diseases Department, Hospital Pourtalès, Neuchâtel, Switzerland, 6 Division of Gastroenterology, Kantonsspital St. Gallen, St. Gallen, Switzerland

* David.Semela@kssg.ch

## Abstract

### Aim

Fatigue is the most commonly reported symptom of Hepatitis C Virus (HCV) infected patients and severely impacts their quality of life. The aim of this study was to measure the impact of 3D regimen treatment on the fatigue, daytime physical activity and sleep efficiency of HCV infected patients with fatigue.

### Methods

HEMATITE was an observational, prospective, open-label, single-arm, Swiss multi-centric study in mono-infected HCV genotype 1 patients. The 28 week observation period comprised of 4 weeks preparation, 12 weeks treatment and 12 weeks follow-up. Fatigue was assessed using the fatigue severity scale (FSS) questionnaire. Patients with FSS $\geq$ 4 (clinically significant fatigue) were included. The activity tracker, ActiGraph GT9X Link®, was used to measure daytime physical activity and sleep efficiency. Outcome analysis was performed on a scaled down intention to treat (sdITT) population, which excluded patients with insufficient tracker data at all study visits and a modified ITT (mITT) population, which consisted of patients with complete tracker data at all study visits.

### Results

Forty of 41 patients in the ITT population had a sustained virologic response 12 weeks post-treatment (SVR12). Mean baseline FSS score was 6.0 for the sdITT population and 5.9 for the mITT population and decreased from baseline to 12 weeks post-treatment by 2.6 (95% confidence interval [CI]: 2.1, 3.1) for the sdITT (n = 37) population and 2.8 (95% CI: 2.2, 3.4)

**Data Availability Statement:** The minimal data are within the paper and its Supporting Information files. For any other data, please contact the corresponding author.

**Funding:** AbbVie Schweiz AG was involved in the study design, data analysis, decision to publish, and preparation of the manuscript. The specific roles of the authors are articulated in the 'author contributions' section.

**Competing interests:** Nasser Semmo has received research grants, consulting fees and/or speaker fees from AbbVie and Gilead and consulting fees from MSD. Beat Müllhaupt has received speaking and/or consulting fees from Merck/MSD, AbbVie, Intercept, Astra, Bayer, BMS, Gilead and research support from Gilead. Lorenzo Magenta has received research grants, consulting fees and/or speaker fees from AbbVie, Gilead, Janssen, BMS and MSD. Olivier Clerc has received consulting fees from AbbVie. David Semela has received research grants, consulting fees and/or speaker fees from AbbVie, Bayer, BMS, Gilead, Intercept and MSD. Ralph Torgler and Lisa Ruckstuhl are employees of AbbVie and own stock/options of AbbVie. This does not alter our adherence to PLOS ONE policies on sharing data and materials.

**Abbreviations:** AEs, adverse events; BMI, body mass index; CI, confidence interval; CNS, central nervous system; FSS, fatigue severity scale; HCV, hepatitis C virus; ITT, intention to treat; mITT, modified intention to treat; SAEs, serious adverse events; sdITT, scaled down intention to treat; SVR12, sustained virologic response 12 weeks post-treatment.

for the mITT (n = 24) population. Mean daytime physical activity or sleep efficiency did not change considerably over the course of the study.

## Conclusion

Measurement by the activity tracker of mean day time physical activity did not show a considerable change from baseline to SVR12 upon treatment with 3D regimen. Nevertheless, a reduction of fatigue as assessed with the validated fatigue severity scale (FSS) was observed, suggesting a causative role of HCV in this extrahepatic manifestation.

## Trial registration

**ClinicalTrials.gov identifier:** NCT03002818.

## Introduction

Chronic hepatitis C virus (HCV) infection, characterized by persistent hepatic inflammation, affects around 71 million people worldwide and can lead to liver fibrosis, cirrhosis and hepatocellular carcinoma [1]. Extrahepatic systems, such as the central nervous system (CNS), are also associated with chronic HCV, and neuropsychiatric disorders are more prevalent among HCV infected patients than the general population [2–4]. The reason for this is manifold: HCV infected patients are more likely to have pre-existing mental illness or to abuse psychoactive substances [5]; cirrhosis may lead to minimal or overt encephalopathies with a broad range of neuropsychological symptoms [6]; pharmaceutical treatment for HCV can exacerbate certain neuropsychiatric symptoms [7]; and HCV itself may have a direct biologic effect on the CNS [8].

HCV infected patients may present with many neuropsychiatric disorders including fatigue, anxiety, depression and cognitive impairment, which can significantly reduce their quality of life [3, 9–12]. Of these, fatigue is the most commonly reported symptom [10, 13, 14] and may occur in conjunction with sleep disturbances [15]. Reduced quality of life in HCV infected patients is independent of liver damage [13]. HCV eradication following treatment with interferon alpha and ribavirin has been shown to improve neurocognitive symptoms in patients [16].

Treatment of HCV has progressed in recent years with the approval of 3D regimen, an interferon-free regimen indicated in HCV genotype 1 infected patients. It has been shown to result in high rates of sustained virologic response 12 weeks post-treatment (SVR12) [17, 18]. The treatment consists of paritaprevir, a protease inhibitor boosted by ritonavir; ombitasvir, a HCV nonstructural protein 5A (NS5A) replication complex inhibitor; and dasabuvir, a non-nucleoside RNA polymerase inhibitor. To date, there has been only few publications on the characterization of fatigue through treatment with newly developed direct-acting antivirals [19], and no research into the effect of 3D regimen on the quality of life of HCV patients suffering from fatigue. A recently published work of Durcan et al. found, that direct antivirals did not lead to depression, anxiety or fatigue and did not decrease liver-specific quality of life [20].

The aim of this observational study, HEMATITE, was to measure the impact of 3D regimen on the daytime physical activity, fatigue and sleep efficiency of HCV patients with fatigue. Patients with predisposing factors of fatigue, such as severe depression, cirrhosis and cancer were excluded from the study.

## Materials and methods

### Study design

HEMATITE was a single-arm, prospective, post-marketing, observational study in HCV patients receiving 3D regimen according to routine clinical practice. The 28 week observation period was comprised of 4 weeks preparation, 12 weeks treatment and 12 weeks follow-up. At Study Visit 1 (Day -28; before treatment start), patient screening took place; at Study Visit 2 (Day 1; treatment start), baseline data were obtained; Study Visit 3 (Day 28) was an interim visit; Study Visit 4 (Day 84) was at the end of treatment; and at Study Visit 5 (Day 168), SVR12 was assessed (S1 Fig). The study enrolled patients over a 12 month period and was conducted at five HCV competence centers in Switzerland from Mar 2017 to Apr 2018: Kantonsspital St. Gallen, St. Gallen; University Hospital Zürich, Zürich; Universitätsspital Bern, Bern; Fondazione Epatocentro Ticino, Lugano and Hôpital Neuchâtelois Pourtalès, Neuchâtel. The protocol was approved by the Ethikkommission Ostschweiz, St. Gallen and conducted in accordance with the Declaration of Helsinki. Written informed consent was obtained from all participants included in the study.

### Patients

Patients diagnosed with HCV, whom the physician had already decided to treat with 3D regimen, were offered the opportunity to participate in the study. Patients were eligible if they were aged 18 years or older, were mono-infected with chronic HCV genotype 1, had fatigue (fatigue severity scale [FSS] $\geq$ 4 [21]), were treatment naïve and did not have liver cirrhosis. Patients were excluded if they had fatigue from sources other than HCV (e.g. severe depression, cancer and hormonal disorders), had conditions that did not allow them to adhere to the protocol or were wheelchair dependent.

### Treatment

Commercially available 3D regimen, i.e. ombitasvir, paritaprevir and ritonavir tablets (Viekirax®, AbbVie Deutschland GmbH & Co. KG, Germany) and dasabuvir tablets (Exviera®, AbbVie Deutschland GmbH & Co. KG, Germany), was used as per routine clinical practice, local label and guidelines. The treatment regimen was at the discretion of the physician and was decided upon prior to offering the patient the opportunity to participate in the study.

### Fatigue severity scale

Fatigue was assessed at baseline (Day 1), Day 28, Day 84 and 12 weeks post-treatment (Day 168) using the FSS questionnaire which has been validated for use in chronic HCV and is an adequate measure of fatigue outcomes in HCV clinical trials [21]. Patients answered nine questions using a scale from one (strongly disagree = 1 point) to seven (strongly agree = 7 points) (S2 Fig). A fatigue score was obtained by calculating the mean score of the nine items. The higher the score (with a maximum of 7), the greater the fatigue. Clinically significant fatigue was defined as a score equal or above four [21, 22].

### Activity tracker

The activity tracker, ActiGraph GT9X Link® (ActiGraph LLC, Pensacola, Florida, USA), was used to measure daytime physical activity and sleep efficiency. ActiGraph GT9X Link® is a Class I medical device within the European Union [23] and ActiGraph® devices have been used in several research studies and clinical trials to measure physical activity, energy expenditure and sleep/wake behavior [24–26]. Patients wore the activity trackers on their non-

dominant arm for 4 weeks prior to each study visit (baseline [Day 1], Day 28, Day 84 and 12 weeks post-treatment [Day 168]). The patients were instructed on the use of the device by study personnel and were reminded when to put it on. The daytime physical activity was recorded by the activity tracker in counts. Counts were the sum of accelerometer values (raw data at 30 Hz) that had passed through a proprietary filtering process to eliminate non-human movement. Sleep efficiency was also recorded by the activity tracker and was defined as the percentage of time scored as sleep during the sleep period. At 12 weeks post-treatment (Day 168), tracker data collected using device specific software (ActiSync) were processed by the biostatistician. The data were not visible to the Investigator or patients during the study.

## Patient characteristics, vital signs and laboratory parameters

Patient data (demographics, disease characteristics, comorbidities, concomitant medication and treatment details) were obtained from the patient's medical records.

Vital signs (blood pressure, pulse, weight and height, body mass index [BMI] and temperature) and laboratory parameters (HCV RNA level, alanine aminotransferase, aspartate aminotransferase, total bilirubin, hemoglobin, creatinine, ferritin, thyroid stimulating hormone, fasting glucose and human chorionic gonadotropin) were measured at each study visit.

## Variables

The primary outcome variable was a change in mean daytime physical activity from baseline (Day 1) to 12 weeks post-treatment (Day 168). The secondary variables were a change in FSS score from baseline (Day 1) to 12 weeks post-treatment (Day 168); a correlation between mean daytime physical activity and FSS score from baseline (Day 1) to 12 weeks post-treatment (Day 168); sleep efficiency at baseline (Day 1), during and after 12 weeks post-treatment (Day 168); and the proportion of patients who achieved SVR12.

## Statistical methods

A sample size of 100 was planned, which would have had 80% power to detect a change from baseline of the effect size 0.29 using a two-sided one-sample t-test with a significance level of 5%. However, the anticipated sample size was not met. The main reasons for this were the launches of new pangenotypic treatment options and the abolition of a limitatio laid down by the Swiss Federal Office of Public Health.

Safety analysis was performed on the intention to treat (ITT) population which comprised all patients who received the study treatment at least once. Outcome analysis was performed on the following two populations: the scaled down ITT (sdITT) population, defined as the ITT population minus patients who discontinued or were excluded because of missing or insufficient tracker data at all study visits, and the modified ITT (mITT) population, defined as patients in the sdITT population who had complete tracker data at all five study visits.

Activity tracker data for 10 working days before each scheduled study visit (baseline [Day 1], Day 28, Day 84 and 12 weeks post-treatment [Day 168]) were used to calculate the mean daytime physical activity and sleep efficiency for each tracker phase of the study. Days missing $\geq$ 2 hours activity data during the daytime (as determined by a built-in wear time sensor) were excluded. Data for these days were replaced by daytime values in a pre-defined order, starting with the most recent day of the preceding week.

Data for each study visit were analyzed using descriptive statistical analysis. For daytime physical activity, FSS and sleep efficiency, the mean change and the 95% confidence interval (CI) between baseline (Day 1) and Day 28, Day 84 and 12 weeks post-treatment (Day 168) was calculated. The difference between mean physical activity at baseline (Day 1) and at 12 weeks post-

treatment (Day 168) was analyzed by a two-sided one-sample t-test with a significance level of $\alpha$ = 0.05. The correlation between mean daytime physical activity and FSS score was analyzed by Spearman's rank correlation coefficient. The proportion of patients who achieved SVR12 and the 95% CI were also calculated as standard Wald intervals using the estimated standard error.

Subgroup analysis was performed to assess the effect of the following factors on daytime physical activity, FSS and sleep efficiency: concomitant ribavirin, gender, fibrosis stage, age and genotype 1 subtype. Differences were determined by t-test or Mann-Whitney-U test with a significance level of $\alpha$ = 0.05. All statistical analysis was performed using SPSS® for Windows, version 22.0.

### Safety analysis

Safety assessments were performed in a standardized method at each study visit and included the evaluation of adverse events (AEs): serious AEs (SAEs), severe AEs, treatment-related AEs, procedure-related AEs (i.e. AEs associated with wearing the activity tracker) and AEs leading to discontinuation. An increase in the FSS score by $\geq$ 1 was also reported as an AE.

## Results

### Patients

The number of patients who participated in the study are shown in **Fig 1**. Forty-five patients were screened, of whom 41 were eligible and received 3D regimen treatment at least once (ITT population). The outcome analysis sdITT population comprised 37 patients and excluded patients who discontinued (n = 1) or had missing tracker data for all study visits (n = 3). The outcome analysis mITT population comprised 24 patients and excluded patients with partial tracker data sets (n = 13).

The patient characteristics at screening are shown in **Table 1**. Patients had been diagnosed with HCV infection an average of 14.0 (± 9.3) years before screening and were HCV treatment naïve. The most reported ($\geq$ 8.0%) conditions at screening were drug abuse (21.9%), asthenia (9.8%), depression (9.8%), hepatic steatosis (9.8%) and hypertension (9.8%). Forty patients completed the 3D regimen treatment.

### Primary and secondary variables

Mean daytime physical activity (measured in day counts) decreased slightly but did not change significantly over the course of the study (**Fig 2**). The mean baseline (Day 1) day counts was 1,469,569 for the sdITT population and 1,513,166 for the mITT population. The change from baseline (Day 1) to 12 weeks post-treatment (Day 168) was -108,266 day counts (95% CI: -235,037, 18,504) for the sdITT population (p = 0.091) and -98,373 day counts (95% CI: -235,667, 38,921) for the mITT population (p = 0.152) (**Fig 2A and 2B**). There was no correlation between daytime physical activity and FSS score over the course of the study as assessed by Spearman's rank correlation coefficient.

The distribution of FSS score from screening (Day -28) to 12 weeks post-treatment (Day 168) is shown in **Fig 3** for both analysis populations. From screening (Day -28) to baseline (Day 1), no change in FSS was observed (**Fig 3A and 3B**). The mean baseline (Day 1) FSS score was 6.0 for the sdITT population and 5.9 for the mITT population. A reduction in fatigue (i.e. a decrease in mean FSS score) was observed during the treatment period and was sustained until 12 weeks post-treatment (SVR12) (**Fig 3A, 3B and 3C**). The FSS score decreased from baseline (Day 1) to 12 weeks post-treatment (Day 168) by 2.6 (95% CI: 2.1, 3.1) for the sdITT population and 2.8 (95% CI: 2.2, 3.4) for the mITT population (**Fig 3D and 3E**).

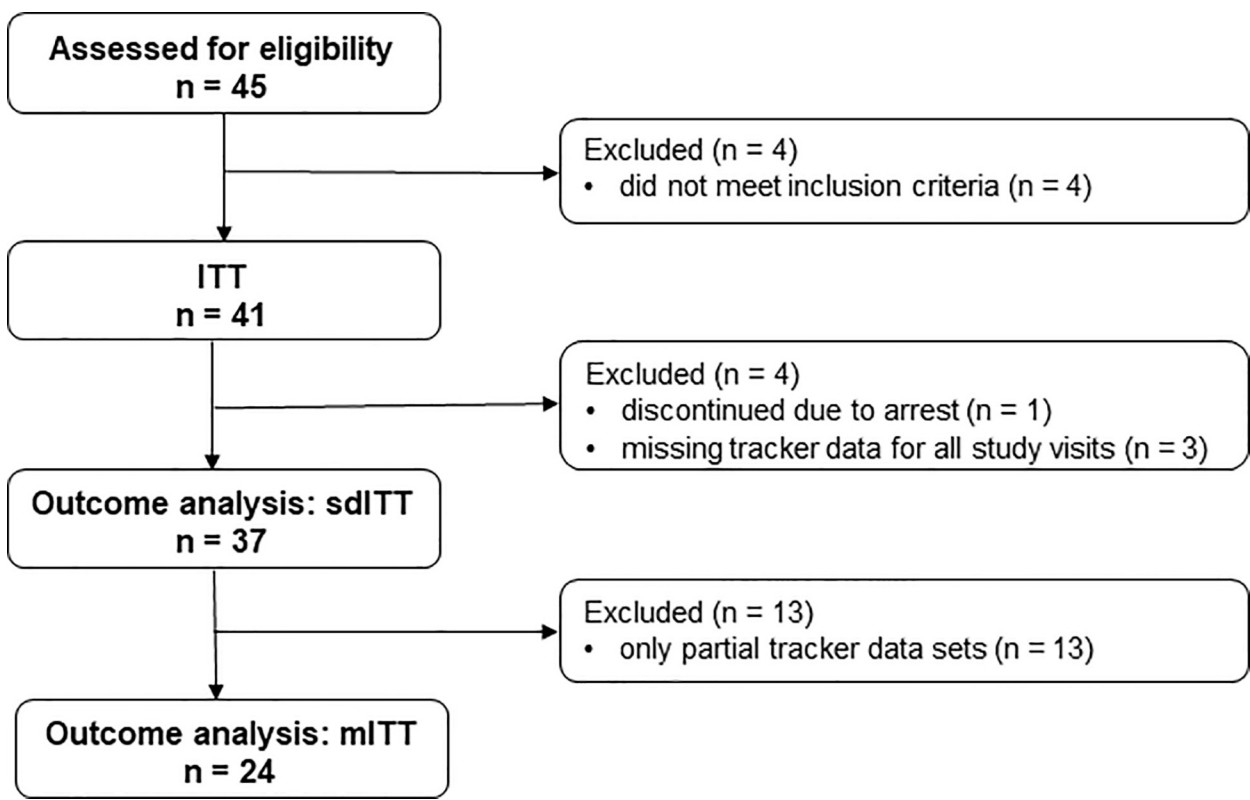

**Fig 1. Flow chart of study participants.** ITT = intention to treat population; sdITT = scaled down ITT population; mITT = modified ITT population.

Similar to daytime physical activity, sleep efficiency decreased slightly but did not change considerably over the course of the study. The mean change in sleep efficiency from baseline (Day 1) to treatment week 4 (day 28) was -0.87% (95% CI: -1.7, -0.03) for the sdITT population and -0.44% (95% CI: -1.5, 0.6) for the mITT population. The mean change in sleep efficiency from baseline (Day 1) to end of treatment (Day 84) was -0.06% (95% CI: -1.0, 0.9) for the sdITT population and 0.01% (95% CI: -1.2, 1.2) for the mITT population. The mean change in sleep efficiency from baseline (Day 1) to 12 weeks post-treatment (Day 168) was -0.6% (95% CI: -0.7, 1.9) for the sdITT population and 0.7% (95% CI: -0.6, 2.1) for the mITT population.

The percentage of patients that reached SVR12 was 97.3% (95% CI: 92.3, 100.0) in the sdITT population and 95.8% (95% CI: 88.3, 100.0) in the mITT population.

### Subgroup analysis

Concomitant ribavirin, gender, fibrosis stage, age and genotype 1 subtype had no effect on daytime physical activity, FSS score or sleep efficiency in the sdITT or mITT populations.

An analysis of several predictors to the three outcome variables daytime physical activity, sleep efficiency and FSS were performed via generalized linear models with repeated measurement. As predictors age (classified by median split: ≤ 50 years/>50 years), gender (male/female), HCV genotype (genotype 1a/genotype 1b), liver fibrosis (yes/no) and ribavirin use (yes/no) were investigated, results are shown for sdITT population (S1 Table) and for mITT population (S2 Table). Overall, for none of the analyzed possible predictive factors the univariate as well as the multivariate analysis showed significance. The corresponding effect sizes demonstrated no or very small effects for these factors. Therefore, the factors age class, sex,

**Table 1. Patient characteristics at screening.**

| Characteristic | n/N | (%) |
|---|---:|---|
| Age [years], mean (± SD) | 49.4 | (± 12.7) |
| ≤ 50 years | 22/41 | (53.7) |
| > 50 years | 19/41 | (46.3) |
| Gender | | |
| Male | 14/41 | (34.1) |
| Female | 27/41 | (65.9) |
| Race | | |
| Caucasian | 41/41 | (100.0) |
| BMI [kg/m$^2$], mean (± SD) | 23.7 | (± 4.4) |
| BMI ≥ 25 (overweight)* | 11/32 | (34.4) |
| Fibrosis stage | | |
| No fibrosis | 15/41 | (36.6) |
| F1 stage | 26/41 | (63.4) |
| Source of HCV infection[†] | | |
| IV drug use | 18/41 | (43.9) |
| Tattoos or piercings | 5/41 | (12.2) |
| Sexual transmission | 3/41 | (7.3) |
| Transfusions | 3/41 | (7.3) |
| Other | 5/41 | (12.2) |
| Unknown | 13/41 | (31.7) |
| HCV genotype | | |
| Genotype 1a | 24/41 | (58.5) |
| Genotype 1b | 17/41 | (41.5) |
| Smoking and alcohol consumption | | |
| Smokers | 21/38 | (55.3) |
| Consume alcohol | 25/39 | (64.1) |
| Laboratory markers, mean (± SD) | | |
| HCV RNA level [log$_{10}$ IU/mL], n = 38 | 6.6 | (± 6.8) |
| AST [U/L], n = 38 | 40.7 | (± 21.0) |
| ALT [U/L], n = 38 | 48.0 | (± 28.2) |
| Total bilirubin [μmol/L], n = 37 | 11.7 | (± 6.3) |
| Hemoglobin [g/L], n = 26 (normal range: 120–180 g/L) | 143.6 | (± 15.0) |
| Creatinine [μmol/L], n = 38 (normal range: 44–106 μmol/L) | 67.3 | (± 12.9) |
| Ferritin [μg/L], n = 22 (normal range: 30–200 μg/L) | 172.4 | (± 130.4) |
| TSH [mU/L], n = 24 (normal range: 0.3–3.5 mU/L) | 1.7 | (± 1.3) |
| Fasting glucose [mmol/L], n = 26 (normal range: 3.9–5.6 mmol/L) | 5.0 | (± 0.6) |

ALT = alanine aminotransferase; AST = aspartate aminotransferase; BMI = body mass index; HCV = hepatitis C;
IV = intravenous; RNA = ribonucleic acid; SD = standard deviation; TSH = thyroid stimulating hormone.

[†] Multiple answers were reported

*BMI data only available for 32 patients

fibrosis, HCV genotype and ribavirin use had no influence of the three outcome variables mean daytime physical activity, sleep efficiency and FSS.

## Safety

During the study, there were no SAEs, procedure-related AEs or AEs that led to discontinuation. Twenty-one (51.2%) of 41 patients experienced an AE during the study. Thirteen

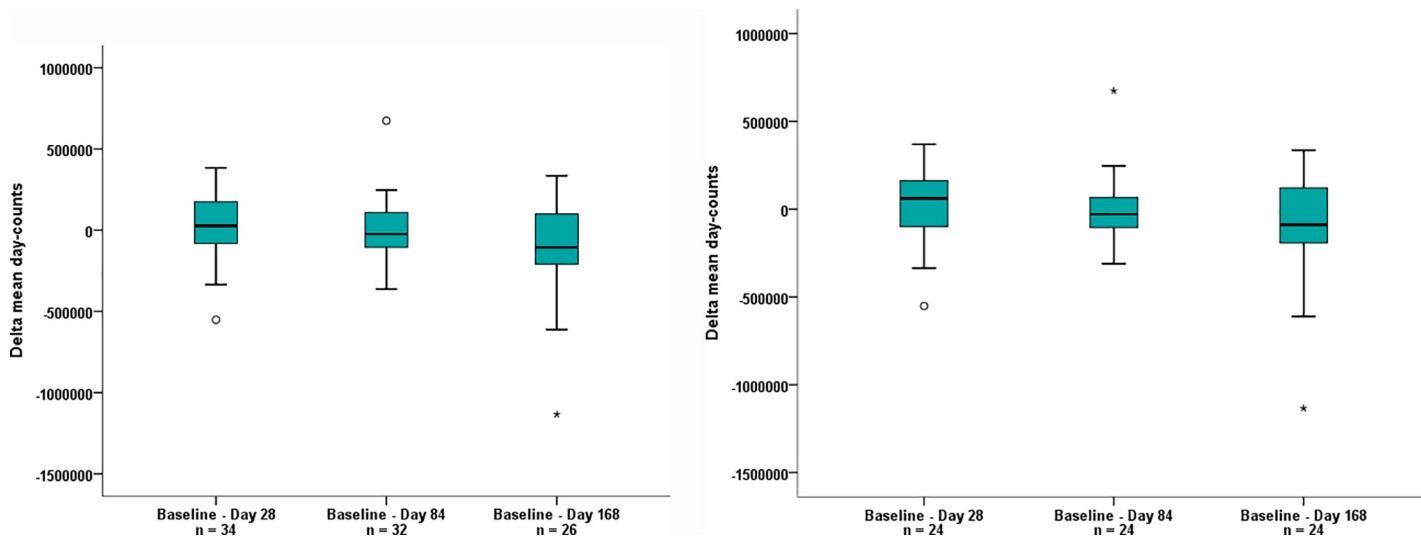

**Fig 2.** Change in mean daytime physical activity from baseline (Day 1) to Day 28, Day 84 and 12 weeks post-treatment (Day 168) is shown for (a) the sdITT population and (b) the mITT population. ° = outlier (value maximum 1.5 to 3 fold box length); * = extreme value (> 3 fold box length); mITT = modified intention to treat; sdITT = scaled down intention to treat; V = study visit.

(31.7%) patients experienced at least one AE that was considered as treatment-related by the Investigator. The most commonly reported AEs and treatment-related AEs are shown in **Table 2**.

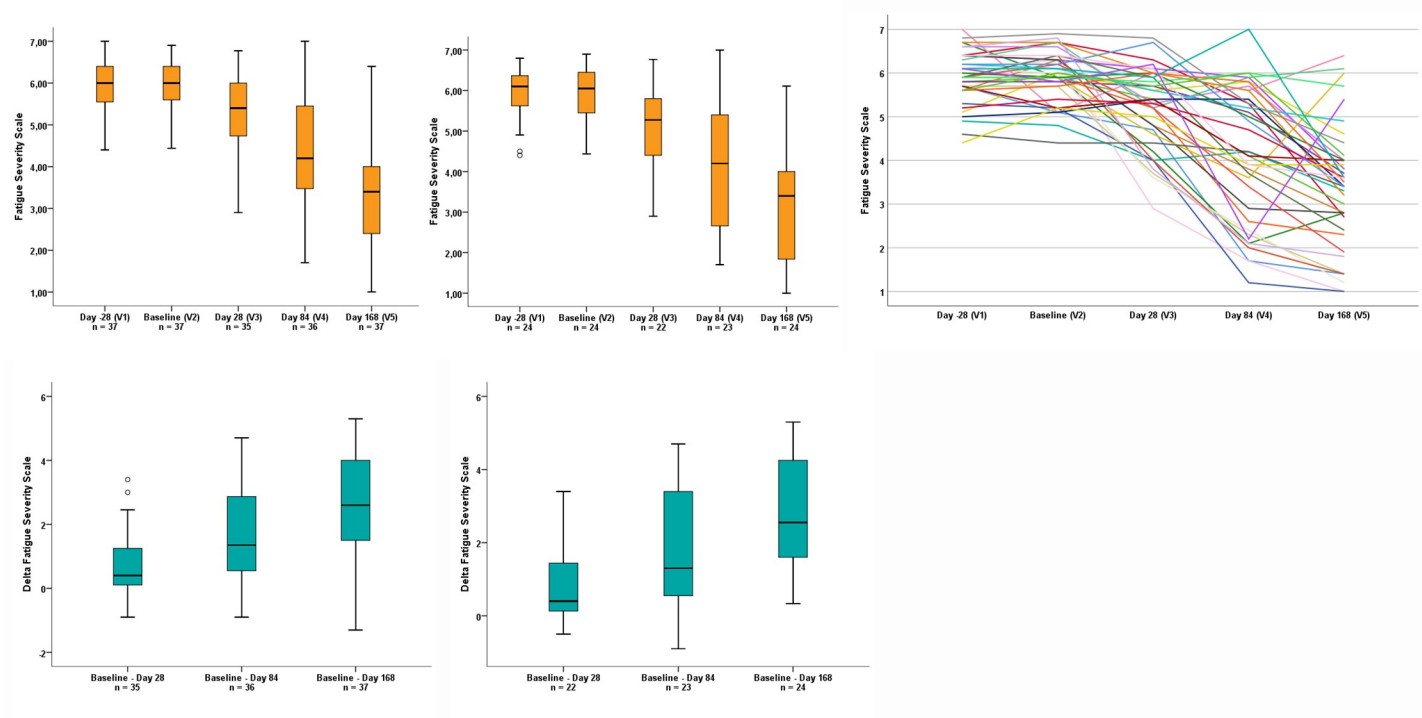

**Fig 3.** Distribution of mean FSS score at screening (Day -28), baseline (Day 1), Day 28, Day 84 and 12 weeks post-treatment (Day 168) is shown for (a) the sdITT population, (b) the mITT population and (c) all patients that had a completed FSS questionnaire for all study visits (n = 39). The decrease in mean FSS score from baseline (Day 1) to Day 28, Day 84 and 12 weeks post-treatment (Day 168) is shown for (d) the sdITT population and (e) the mITT population. ° = outlier (value maximum 1.5 to 3 fold box length); FSS = fatigue severity scale, mITT = modified intention to treat; sdITT = scaled down intention to treat; V = study visit.

**Table 2. Adverse events and treatment-related adverse events occurring in ≥ 2.5% of patients (N = 41).**

| Adverse event [†] | n | (%) |
|---|---|---|
| **Any AE [‡]** | **21** | **(51.2)** |
| **Any SAE** | **0** | **(0)** |
| **AEs leading to discontinuation** | **0** | **(0)** |
| **AEs occurring in ≥ 2 patients** | | |
| Nausea | 6 | (14.6) |
| Dizziness / vertigo | 3 | (7.3) |
| Anemia | 2 | (4.9) |
| Abdominal pain | 2 | (4.9) |
| Emesis | 2 | (4.9) |
| Epigastric pressure pain | 2 | (4.9) |
| Flu like symptoms | 2 | (4.9) |
| FSS increased ≥ 1 | 2 | (4.9) |
| Total bilirubin increase | 2 | (4.9) |
| **Any treatment-related AE** | **13** | **(31.7)** |
| **Treatment-related AEs occurring in ≥ 2 patients** | | |
| Nausea | 4 | (9.8) |
| Total bilirubin increase | 2 | (4.9) |
| Anemia | 2 | (4.9) |
| FSS increased ≥ 1 | 2 | (4.9) |

AE = adverse event; FSS = fatigue severity scale: SAE = serious adverse event.

[†] Fatigue is not included as it was part of the inclusion criteria. An increase of ≥ 1 point in the FSS was documented as an AE.

[‡] Any AE occurred in 21 out of 41 patients (51.2%).

## Discussion

This observational study examined the impact of treatment with 3D regimen on HCV patients suffering from fatigue. Fatigue is experienced by over half of HCV patients [10, 13, 14] and is one of a number of extrahepatic manifestations responsible for their reduced quality of life [2, 27]. In our study, patient fatigue was independent of gender, age and genotype 1 subtype. In addition, patient fatigue seems to be independent of fibrosis stage, although our observation is based on F0 and F1 patients exclusively. Moreover, fatigue could not be attributed to cirrhosis, cancer, severe depression or prior-HCV treatment, as patients with these predisposing factors were excluded from the study. This would indicate that HCV infection itself is most likely responsible for fatigue; a finding that supports research by others [9, 28]. While it is not known exactly how HCV causes fatigue and other neuropsychiatric disorders, it is thought that the virus penetrates the CNS and causes neuroinflammation which results in alterations in cerebral metabolism, immune activation and neurotransmission [29].

In this study, treatment with 3D regimen reduced fatigue over time in non-cirrhotic, treatment naïve patients with HCV genotype 1 infection. A reduction in mean FSS score of 2.8 (95% CI: 2.21, 3.43) from baseline (Day 1) to 12 weeks post-treatment (Day 168) was observed in the mITT population. This change was significant with $p < 0.001$, Friedman's ANOVA. This was a notable decrease considering that a mean FSS score difference of ≥ 0.7 is considered clinically important [22]. All but one patient treated with 3D regimen achieved viral elimination as determined by undetectable HCV RNA 12 weeks post-treatment (SVR12) resulting in an SVR12 rate of > 97%. This high SVR12 achieved in a real life setting confirms previous results from registered trials with 3D regimen [17, 30, 31]. In the patient who did not achieve

SVR12, the FSS score decreased from baseline (Day 1) until end of treatment (Day 84) but increased again at 12 weeks post-treatment (Day 168). There was no virology data available for this patient at the end of treatment visit to determine if a corresponding decline in HCV RNA occurred at this time point.

The accuracy of the validated FSS questionnaire was supported by the fact that no change in FSS score was observed from screening (Day -28) to baseline (Day 1); FSS score only decreased over the treatment period and was sustained until 12 weeks post-treatment (SVR12).

Interestingly, fatigue reduction did not correlate with an increase in mean daytime physical activity during the working week. Indeed, mean daytime activity during the working week did not change during the course of treatment. This result may be partly explained by the fact that a high proportion of jobs are sedentary and most working conditions do not allow for large changes in physical activity. Similarly, sleep efficiency assessed by an activity tracker did not improve over the course of the study. This is in line with the findings of Heeren et al., where there was no correlation between the results of a fatigue questionnaire (fatigue impact scale) and sleep actigraphy data in HCV patients observed and the authors suggested that bad sleep quality may therefore not be associated with increased nocturnal motor activity [24].

This is the first study to use an activity tracker in HCV patients during a course of treatment. While activity trackers are useful tools in monitoring physical activity and sedentary behavior over time, these devices have some limitations. The accuracy of some activity trackers is variable and they may over- or underestimate total activity [32]. Differences in physical activity may be too small to be detected by the activity tracker and studies have shown that they may not accurately detect light physical activity such as washing dishes, cooking food and walking slowly [33]. For a device to accurately report on changes in daily activity, it must be worn consistently [34]. This study had a particularly long tracking period which may have led to study fatigue and inconsistent wearing of the tracking device resulting in missing data and therefore data that were not representative of actual physical activity. Furthermore, wearing a tracker may trigger physical activity in patients per se and thus may confound routine activity. The relatively small patient number and the inclusion of only F0 and F1 patients further add to the limitations of our study.

Differentiating between HCV-associated and non-associated fatigue, physical inactivity and comorbidities is difficult. In our study, 34.4% of patients were overweight (BMI $\geq$ 25), 64.1% consumed alcohol and 55.3% were smokers at screening. These factors may have contributed to fatigue and a lack of physical activity in patients during the study. In addition, there were concomitant medications that may have exacerbated some of the neuropsychiatric symptoms of HCV. In patients receiving concomitant sedative medication, there was no particular trend in FSS score over the course of the study (S3 Fig). Interestingly, despite its hemolytic characteristics and thus potential to cause anemia [35], co-administration of ribavirin did not impact fatigue or physical activity. This is in line with previous reports characterizing health related quality of life during co-administration of ribavirin and other direct-acting antivirals [36].

3D regimen was well-tolerated and the AE profile was in line with that previously reported [17, 30]. In conclusion, treatment with 3D regimen reduced fatigue in treatment naïve genotype 1 mono-infected HCV patients over time. Thus, successful treatment may reduce the physical burden of manifestations of HCV and improve the overall quality of life of patients.

## Supporting information

**S1 Checklist.**
(PDF)

**S1 Fig. Study design and schedule of assessments.** The 28 week observation period was comprised of 4 weeks preparation, 12 weeks treatment and 12 weeks follow-up. Screening took place at Day -28 (Study Visit 1). Fatigue was assessed by FSS questionnaire on Days 1, 28, 84 and 168 (Study Visits 2, 3, 4 and 5, respectively). Daytime physical activity and sleep efficiency were assessed by activity tracker during 4 x 4 week tracker phases. Baseline data were collected in tracker phase 1. A data set of 2 eligible weeks (10 working days) was used to assess daytime physical activity and sleep efficiency from tracker phases 1, 2, 3 and 4. SVR12 was determined on Day 168 (12 weeks post-treatment).
(TIF)

**S2 Fig. Fatigue severity scale.** Taken from *Krupp LB, LaRocca NG, Muir-Nash J, Steinberg AD. The fatigue severity scale. Application to patients with multiple sclerosis and systemic lupus erythematosus. Arch Neurol 1989;46:1121–1123.*
(TIF)

**S3 Fig. Distribution of mean FSS score at screening (Day -28), baseline (Day 1), Day 28, Day 84 and 12 weeks post-treatment (Day 168) is shown for all patients receiving sedative medication (n = 11).**
(TIF)

**S1 Table. Possible predictors for the outcomes, sdITT (n = 37).**
(DOC)

**S2 Table. Possible predictors for the outcomes, mITT (n = 24).**
(DOC)

**S1 File.**
(PDF)

## Author Contributions

**Conceptualization:** Lisa Ruckstuhl, Ralph Torgler, David Semela.

**Data curation:** Lisa Ruckstuhl, Ralph Torgler.

**Formal analysis:** Lisa Ruckstuhl, Ralph Torgler.

**Investigation:** Nasser Semmo, Beat Müllhaupt, Lorenzo Magenta, Olivier Clerc, David Semela.

**Methodology:** Lisa Ruckstuhl, Ralph Torgler.

**Project administration:** Lisa Ruckstuhl.

**Supervision:** Ralph Torgler.

**Writing – review & editing:** Nasser Semmo, Beat Müllhaupt, Lisa Ruckstuhl, Lorenzo Magenta, Olivier Clerc, Ralph Torgler, David Semela.

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
