## [Decision Letter · Decision Letter 0]

5 May 2020

PONE-D-20-05447

A prospective, multicenter, post-marketing observational study to measure the quality of life of HCV genotype 1 infected, treatment naïve patients suffering from fatigue and receiving 3D regimen: the HEMATITE study.

PLOS ONE

Dear Ruckstuhl,

Thank you for submitting your manuscript to PLOS ONE. After careful consideration, we feel that it has merit but does not fully meet PLOS ONE’s publication criteria as it currently stands. Therefore, we invite you to submit a revised version of the manuscript that addresses the points raised during the review process.

There are a number of issues to be revised. Please pay special attention to:

- Improve the quality of the figures

- Update references about evolution of quality of life after hepatitis C treatment.

We would appreciate receiving your revised manuscript by Jun 19 2020 11:59PM. To enhance the reproducibility of your results, we recommend that if applicable you deposit your laboratory protocols in protocols.io, where a protocol can be assigned its own identifier (DOI) such that it can be cited independently in the future. For instructions see: http://journals.plos.org/plosone/s/submission-guidelines#loc-laboratory-protocols

We look forward to receiving your revised manuscript.

Kind regards,

Jose Ignacio Herrero

Academic Editor

PLOS ONE

Journal Requirements:

2. Please ensure you have included the registration number for the clinical trial referenced in the manuscript.

'Nasser Semmo has received research grants, consulting fees and/or speaker fees

from AbbVie and Gilead and consulting fees from MSD. Beat Müllhaupt has received

speaking and/or consulting fees from Merck/MSD, AbbVie, Intercept, Astra, Bayer,

BMS, Gilead and research support from Gilead. Lorenzo Magenta has received

research grants, consulting fees and/or speaker fees from AbbVie, Gilead, Janssen,

BMS and MSD. Olivier Clerc has received consulting fees from AbbVie. David Semela

has received research grants, consulting fees and/or speaker fees from AbbVie, Bayer,

BMS, Gilead, Intercept and MSD. Ralph Torgler and Lisa Ruckstuhl are employees of

AbbVie and own stock/options.'  

We note that one or more of the authors are employed by a commercial company: AbbVie.

Additional Editor Comments (if provided):

The authors have studied the evolution of the quality of life (sleep efficiency, activity and fatigue) in a group of 41 patients with hepatitis C treated with the 3d combo (they expected to recruit 100). They have found that the treatment was followed by an improvement in the fatigue scale. This is an interesting issue, but there a number of issues that should be revised. Most of them have been mentioned by the reviewers.

- The conclusion of the abstract about the proportion of SVR should be eliminated. This is not an aim of the study.

- The references about the topic should be updated (not only the reference suggested by Reviewer 2).

- Quality of the figures should be improved.

- Fatigue severity score should be better explained in the methods section. It seems that 7 is the worst score. Is it true? Do the patients have a score of 6 at baseline? I think this is unlikely (all of them are F0/F1 patients). Could the authors give a reference value from general population?

- Could the authors give some information about day count in general population?

- Table 1 should be revised: AST, ALT, bilirubin levels are expressed in an uncommon way (it is more frequent to give them as median & IQR. HCV RNA level should be expressed as log. Hemoglobin, ferritin, TSH, and glucose values are missing from a high proportion of patients.

Reviewers' comments:

Reviewer's Responses to Questions

**Comments to the Author**

1. Is the manuscript technically sound, and do the data support the conclusions?

Reviewer #1: Yes

Reviewer #2: Yes

Reviewer #3: Yes

Reviewer #4: Yes

2. Has the statistical analysis been performed appropriately and rigorously? 

Reviewer #1: Yes

Reviewer #2: Yes

Reviewer #3: Yes

Reviewer #4: Yes

3. Have the authors made all data underlying the findings in their manuscript fully available?

Reviewer #1: No

Reviewer #2: Yes

Reviewer #3: Yes

Reviewer #4: Yes

4. Is the manuscript presented in an intelligible fashion and written in standard English?

Reviewer #1: Yes

Reviewer #2: Yes

Reviewer #3: Yes

Reviewer #4: Yes

5. Review Comments to the Author

Reviewer #1: The authors conducted a multicenter, prospective observational study to measure the impact of 3D regimen on the fatigue, daytime physical activity and sleep efficiency of genotype 1 HCV mono-infected patients. Overall the manuscript was well-written and understandable.

I would like to highlight some points as follows:

Major Comments:

1) Data availability statement reflects data are fully available without restriction. However, I can´t see the study database. Is it fully available? If your data is only available upon request, explain it in the Data Availability Statement, as it will be published in the article.

2) One of the secondary variables analyzed is the sleep efficiency at baseline, during treatment and at day 168. In Lines 215-217 is presented the mean change from baseline to Day 168. Please, include results of sleep efficiency change from baseline to Day 28 and from baseline to Day 84, to see the treatment impact on sleep efficiency. Discuss the results in the Discussion Section.

3) Subgroup analysis: it is said that RBV use, gender, liver fibrosis and genotype 1 subtype had no effect on the different outcomes analyzed. The results of this analysis should be presented in a Table, or at least as Supplementary Material.

Minor Comments:

1) The resolution of Fig 2 and 3 is low and difficult to understand.

2) Characteristic of patients as mono-infected HCV is well described in the Methods Section. However, it should be clarified in other parts of the manuscript: Abstract (methods) and Discussion (line 288).

3) Exclusion criteria should be reported in the Abstract, if length allows it.

4) Clarify if informed consent was a written informed consent (line 78).

5) Table 1: BMI data only available in 32 patients could be reflected in the legend, or indicating n=32 as in the laboratory markers.

6) Line 194: Colud you indicate the Spearman´s rank correlation coefficient value? Is it 0?

7) Table 2 Legend: It is referred that data of "Any AE" is not available for all ITT population. Please, indicate in how many patients you have this information. Percentage is calculated taking into account all the 41 patients. Please, calculate this percentage for the "n" patients you have "Any AE" data.

8) Line 234: indicate in parentheses that fibrosis stage is (F0-F1).

9) Discussion section: it should be recognized as a limitation that no F2-F3 patients could be included in the study.

10) Other articles in the literature have found RBV does not impact on HRQoL. References could be included.

11) Corresponding author is not the same in the Title Page and the Submitted form.

Reviewer #2: Please provide a better images of figures. It is impossible to read numbers.

Authors can give the statistical data about the change in the fatigue parameters and comment on the significance of this reduction.

It is not true that fatigue was not investigated earlier in this field (line 59 “ To date, there has been no research into the effect of 3D regimen on the quality of life of HCV patients suffering from fatigue.”) because there is already a published paper about this.( Durcan E, Hatemi I, Sonsuz A, Canbakan B, Ozdemir S, Tuncer M. The effect of direct antiviral treatment on the depression, anxiety, fatigue and quality-of-life in chronic hepatitis C patients. Eur J Gastroenterol Hepatol. 2020 Feb; 32(2): 246-250 doi:10.1097/MEG.0000000000001501. PubMed PMID: 31441798.)

The fatigue is a common symptom in chronic liver disease even if it is not related to viral infection for example primary biliary cholangitis, because of that it is not sense to say “ Reduced quality of life in HCV infected patients is independent of liver damage, indicating that the virus itself is responsible.” The virus is not responsible, chronic liver disease is responsible of reduced quality of life.

Reviewer #3: An observational single-arm study aimed to measure the impact of 3D regimen treatment on fatigue in Hepatitis C Virus infected patients (n=41). Nearly all patients maintained a virologic response to treatment at 12 weeks, and compared to baseline the fatigue severity scale score decreased at 12 weeks. No changes were observed in daytime physical activity or sleep efficiency. The manuscript was clearly written.

Minor revisions:

1- Line 124: Modify the sentence for clarity. “The primary outcome variable was a change in mean daytime physical activity….”

2- Line 155: Indicate the statistical method used to calculate the 95% CI.

3- Paragraph beginning at line 156: Consider building models to predict daytime physical activity, FSS and sleep efficiency.

4- Line 161: Indicate if adverse events were collected according to a standardized method.

5- Line 192: Intraclass correlation coefficients may be superior to Spearman rank correlation coefficients due to repeated measures.

6- Line 221 states, "Concomitant ribavirin, gender, fibrosis stage, age and genotype 1 subtype had no effect on daytime physical activity, FSS score or sleep efficiency in the sdITT or mITT populations." The conclusion in this sentence cannot be supported by results from t-tests or Mann-Whitney-U tests. See comment 3.

Reviewer #4: Thank you very much for your paper. The topic is very important and is also a good investigation. I suggest it might need revision before being accepted. 1.The study design section is confusing: it is not clear when the study visit 1 took place. 2.The title can be revised as "...... observational study to measure the fatigue....", since the study did not measure the quality of life but measured only fatigue. 3.The small sample size can be mentioned as a limitation of the study.

Overall, the study is interesting and I enjoyed reading it.

6. PLOS authors have the option to publish the peer review history of their article (what does this mean?). If published, this will include your full peer review and any attached files.

Reviewer #1: Yes: Regina Juanbeltz

Reviewer #2: No

Reviewer #3: No

Reviewer #4: Yes: Tatevik Balayan

---

## [Author Response · Author response to Decision Letter 0]

30 Jul 2020

PONE-D-20-05447 – Response to Reviewers

Dear Reviewers,

We would like to thank you for the detailed review of our manuscript entitled “HEMATITE Study: A prospective, multicenter, post-marketing observational study to measure the quality of life of HCV genotype 1 infected, treatment naïve patients suffering from fatigue and receiving 3D regimen.” Your comments helped to improve the manuscript. 

Here, we provide a detailed point-by-point reply letter to all reviewers’ comments. We implemented all comments of the reviewers in the revised manuscript. Please find our answers below. The line numbers mentioned correspond to the manuscript version with Track Changes.

We strongly believe that this study will be of significant interest for the readership of PLOS ONE. Thank you for considering this manuscript for publication in PLOS ONE. 

Sincerely,

David Semela, MD PhD & Lisa Ruckstuhl, on behalf of all the authors

Point-by-point reply

Journal Requirements:

Answer: The PLOS ONE’s style requirements have been adjusted.

2) Please ensure you have included the registration number for the clinical trial referenced in the manuscript. 

Answer: The clinical trial registration number (ClinicalTrials.gov Identifier: NCT03002818) is included in the manuscript (line 513).

3) We note that you have included the phrase “data not shown” in your manuscript. Unfortunately, this does not meet our data sharing requirements. 

Answer: We agree with your comment – accordingly a new supplementary figure (supporting information Fig S3) showing the mentioned data was included (line 364, and Fig S3).

4) Thank you for stating the following in the Competing Interests section:

'Nasser Semmo has received research grants, consulting fees and/or speaker fees

from AbbVie and Gilead and consulting fees from MSD. Beat Müllhaupt has received

speaking and/or consulting fees from Merck/MSD, AbbVie, Intercept, Astra, Bayer,

BMS, Gilead and research support from Gilead. Lorenzo Magenta has received

research grants, consulting fees and/or speaker fees from AbbVie, Gilead, Janssen,

BMS and MSD. Olivier Clerc has received consulting fees from AbbVie. David Semela

has received research grants, consulting fees and/or speaker fees from AbbVie, Bayer,

BMS, Gilead, Intercept and MSD. Ralph Torgler and Lisa Ruckstuhl are employees of

AbbVie and own stock/options.' 

Answer: This information is included under the “Competing Interest Section” (line 491).

We note that one or more of the authors are employed by a commercial company: AbbVie.

1) Please provide an amended Funding Statement declaring this commercial affiliation, as well as a statement regarding the Role of Funders in your study. If the funding organization did not play a role in the study design, data collection and analysis, decision to publish, or preparation of the manuscript and only provided financial support in the form of authors' salaries and/or research materials, please review your statements relating to the author contributions, and ensure you have specifically and accurately indicated the role(s) that these authors had in your study. 

Answer: Not applicable, see also below.

Answer: Not applicable, see also below.

Answer: Ralph Torgler and Lisa Ruckstuhl did play a role in the study design, data analysis, decision to publish, and preparation of the manuscript. The Conflict of interest statement has been amended accordingly in the manuscript (lines 491-499). Further, the Funding statement has been added (lines 501-503). Please note, that the funding is also declared in the Acknowledgements. 

2) Please also provide an updated Competing Interests Statement declaring this commercial affiliation along with any other relevant declarations relating to employment, consultancy, patents, products in development, or marketed products, etc. 

Answer: Ralph Torgler and Lisa Ruckstuhl did play a role in the study design, data analysis, decision to publish, and preparation of the manuscript. The Conflict of interest statement has been amended accordingly in the manuscript (lines 491-499). Further, the Funding statement has been added (lines 501-503). Please note, that the funding is also declared in the Acknowledgements. 

Within your Competing Interests Statement, please confirm that this commercial affiliation does not alter your adherence to all PLOS ONE policies on sharing data and materials by including the following statement: "This does not alter our adherence to PLOS ONE policies on sharing data and materials.” 

Answer: We absolutely agree and as requested have added the statement "This does not alter our adherence to PLOS ONE policies on sharing data and materials.” (lines 498/499).

Answer: An updated Funding Statement and Competing Interests Statement was included in our cover letter. 

Additional Editor Comments (if provided):

The authors have studied the evolution of the quality of life (sleep efficiency, activity and fatigue) in a group of 41 patients with hepatitis C treated with the 3d combo (they expected to recruit 100). They have found that the treatment was followed by an improvement in the fatigue scale. This is an interesting issue, but there a number of issues that should be revised. Most of them have been mentioned by the reviewers.

- The conclusion of the abstract about the proportion of SVR should be eliminated. This is not an aim of the study. 

Answer: We agree with this comment and have adjusted the abstract’s conclusion accordingly. 

- The references about the topic should be updated (not only the reference suggested by Reviewer 2). 

Answer: In addition to references suggested by the other reviewers we have added relevant studies in the field of HCV and fatigue: Golabi P, Sayiner M, Bush H, Gerber LH, Younossi ZM. Patient-Reported Outcomes and Fatigue in Patients with Chronic Hepatitis C Infection (Clin Liver Dis. 2017) and Durcan E, Hatemi I, Sonsuz A, Canbakan B, Ozdemir S, Tuncer M. The effect of direct antiviral treatment on the depression, anxiety, fatigue and quality-of-life in chronic hepatitis C patients (Eur J Gastroenterol Hepatol. 2020) (lines 66 - 71).

- Quality of the figures should be improved. 

Answer: Figure quality and resolution have been improved. 

- Fatigue severity score should be better explained in the methods section. It seems that 7 is the worst score. Is it true? 

Answer: Yes, 7 is the worst score. We have clarified this in the Methods section (lines 111-115). In addition, supplementary figure 2 shows the detailed FSS. 

Do the patients have a score of 6 at baseline? I think this is unlikely (all of them are F0/F1 patients). Could the authors give a reference value from general population? 

Answer: It is correct that patients had a score of 6 at baseline (boxplots figures 3a and 3b). The HEMATITE study included only subjects with severe fatigue (defined by an FSS score of ≥4), as described in the ‘materials & methods and patients’ section. We agree, that many patients with F0/F1 will have no or minimal fatigue. Only patients with severe fatigue were selected for this study. Data on FSS from the general population show that healthy subjects reported an FSS of 3.00 ± 1.08, n=454) (Valko PO et al “Validation of the Fatigue Severity Scale in a Swiss Cohort”, Sleep 2008, https://www.ncbi.nlm.nih.gov/pmc/articles/PMC2579971/).

Could the authors give some information about day count in general population? 

Answer: Reference 22 (now Ref 24), Heeren et al., also included healthy subjects, which did not show different day counts (24-h activity level) compared to HCV patients. Such kind of activity data depend strongly on the setting, patient population and the device used (in contrast to our study, Heeren et al. used two different activity monitors) and thus, a direct comparison with our data is difficult (selected fatigue patients, different device). In the HEMATITE study, we only compared baseline activity against SVR12 activity in the same patient; no healthy subjects were included. 

Table 1 should be revised: 

AST, ALT, bilirubin levels are expressed in an uncommon way (it is more frequent to give them as median & IQR). 

Answer: Thank you for this input - we have changed LFT levels expression to mean ±SD. 

HCV RNA level should be expressed as log. 

Answer: We have adjusted HCV RNA levels to log scale.

Hemoglobin, ferritin, TSH, and glucose values are missing from a high proportion of patients. 

Answer: We agree with this remark; as the study setting was observational and non-interventional, the collection of these lab values at each visit was recommended but ultimately at the investigator’s discretion of the different study sites. 

Reviewer #1: 

The authors conducted a multicenter, prospective observational study to measure the impact of 3D regimen on the fatigue, daytime physical activity and sleep efficiency of genotype 1 HCV mono-infected patients. Overall the manuscript was well-written and understandable.

I would like to highlight some points as follows:

Major Comments:

1) Data availability statement reflects data are fully available without restriction. However, I can´t see the study database. Is it fully available? If your data is only available upon request, explain it in the Data Availability Statement, as it will be published in the article. 

Answer: A data availability statement added: Data is available upon request (line 506).

2) One of the secondary variables analyzed is the sleep efficiency at baseline, during treatment and at day 168. In Lines 215-217 is presented the mean change from baseline to Day 168. Please, include results of sleep efficiency change from baseline to Day 28 and from baseline to Day 84, to see the treatment impact on sleep efficiency. Discuss the results in the Discussion Section. 

Answer: As requested we have added data on sleep efficiency in the result section (lines 258-263): 

“The mean change in sleep efficiency from baseline (Day 1) to treatment week 4 (day 28) was -0.87% (95% CI: -1.7, -0.03) for the sdITT population and -0.44% (95% CI: -1.5, 0.6) for the mITT population. The mean change in sleep efficiency from baseline (Day 1) to end of treatment (Day 84) was -0.06% (95% CI: -1.0, 0.9) for the sdITT population and 0.01% (95% CI: -1.2, 1.2) for the mITT population.”

As requested we have adjusted the discussion accordingly added data on sleep efficiency in the result section (lines 337-340):

“This is in line with the findings of Heeren et al., where there was no correlation between the results of a fatigue questionnaire (fatigue impact scale) and sleep actigraphy data in HCV patients observed and the authors suggested that bad sleep quality may therefore not be associated with increased nocturnal motor activity.”

3) Subgroup analysis: it is said that RBV use, gender, liver fibrosis and genotype 1 subtype had no effect on the different outcomes analyzed. The results of this analysis should be presented in a Table, or at least as Supplementary Material. 

Answer: An analysis of several predictors to the three outcome variables daytime physical activity, sleep efficiency and FSS were performed via generalized linear models with repeated measurement. These additional results are now described and presented in supplementary tables (S1 and S2), (lines 270-282):

“An analysis of several predictors to the three outcome variables daytime physical activity, sleep efficiency and FSS were performed via generalized linear models with repeated measurement. As predictors age (classified by median split: ≤ 50 years/>50 years), gender (male/female), HCV genotype (genotype 1a/genotype 1b), liver fibrosis (yes/no) and ribavirin use (yes/no) were investigated, results are shown for sdITT population (supporting information S1 Table) and for mITT population (supporting information S2 Table). Overall, for none of the analyzed possible predictive factors the univariate as well as the multivariate analysis showed significance. The corresponding effect sizes demonstrated no or very small effects for these factors. Therefore, the factors age class, sex, fibrosis, HCV genotype and ribavirin use had no influence of the three outcome variables mean daytime physical activity, sleep efficiency and FSS.”

Minor Comments:

1) The resolution of Fig 2 and 3 is low and difficult to understand. 

Answer: Resolution of figures 2 & 3 has been improved. 

2) Characteristic of patients as mono-infected HCV is well described in the Methods Section. However, it should be clarified in other parts of the manuscript: Abstract (methods) and Discussion (line 288). 

Answer: Done (lines 22 & 370)

3) Exclusion criteria should be reported in the Abstract, if length allows it. 

Answer: Listing the exclusion criteria in the abstract would exceed the max. allowed words (300). We have therefore decided to keep the exclusion criteria in the methods section. 

4) Clarify if informed consent was a written informed consent (line 78). 

Answer: Done (line 90).

5) Table 1: BMI data only available in 32 patients could be reflected in the legend or indicating n=32 as in the laboratory markers. 

Answer: We agree and have added according asterisks/legend.

6) Line 194: Could you indicate the Spearman´s rank correlation coefficient value? Is it 0?

Answer: Please find in the following table the correlation coefficients of Fatigue Severity Scale and mean daytime physical activity. 

The authors find the statement in the manuscript sufficient (“There was no correlation between daytime physical activity and FSS score over the course of the study as assessed by Spearman’s rank correlation coefficient.”).

Population Visit n rSpearman* Interpretation

Scale down IT Baseline (V2) 35 0.300 Fair agreement

 Day 28 (V3) 33 0.131 Poor agreement

 Day 84 (V4) 33 -0.022 Poor agreement

 Day 168 (V5) 28 0.195 Poor agreement

Subgroup 1, scale down ITT Baseline (V2) 10 0,468 Moderate agreement

 Day 28 (V3) 10 0,170 Poor agreement

 Day 84 (V4) 9 -0,268 Fair agreement

 Day 168 (V5) 6 -0,493 Moderate agreement

Subgroup 2, scale down ITT Baseline (V2) 13 0,220 Fair agreement

 Day 28 (V3) 13 0,203 Fair agreement

 Day 84 (V4) 12 0,473 Moderate agreement

 Day 168 (V5) 9 0,200 Fair agreement

Subgroup 3, scale down ITT Baseline (V2) 12 0,025 Poor agreement

 Day 28 (V3) 9 0,444 Moderate agreement

 Day 84 (V4) 10 0,176 Fair agreement

 Day 168 (V5) 11 0,555 Moderate agreement

* Spearman correlation coefficient 

Alternatively according to Bland and Altmann (1995) a multiple regression with subjects as a factor was performed to analyze whether the change of mean daytime physical activity in one subject during the study is paralleled by the change of FSS during the study resulting in the following correlation coefficients: sdITT = 0.1477

mITT = 0.1961.

Reference: Bland JM, Altman DG (1995). Calculating correlation coefficients with repeated observations. BMJ. 310: 446

The ICC, addressed by the reviewer, seems to be more appropriate for single variables measured by more than one investigator (e.g. raters) or variables within the same class which is not applicable in the HEMATITE study.

7) Table 2 Legend: It is referred that data of "Any AE" is not available for all ITT population. Please, indicate in how many patients you have this information. Percentage is calculated taking into account all the 41 patients. Please, calculate this percentage for the "n" patients you have "Any AE" data. 

Answer: We have added the requested information (see Table 2 legend). We have collected the AE data (occurrence of AE or not) in all 41 patients (ITT population).

8) Line 234: indicate in parentheses that fibrosis stage is (F0-F1). 

Answer: We have indicated the fibrosis stage as requested (lines 299-301) and added a phrase in the discussion: “The relatively small patient number and the inclusion of only F0 and F1 patients further add to the limitations of our study.” (lines 356-358).

9) Discussion section: it should be recognized as a limitation that no F2-F3 patients could be included in the study. 

Answer: We agree with this comment. This fact has been considered in the phrase mentioned in comment/answer 8 (see above, lines 356-358). Please note that the inclusion of only F0/F1 patients does not minor the value of the observation as we exclusively included patients suffering from severe fatigue.

10) Other articles in the literature have found RBV does not impact on HRQoL. References could be included. 

Answer: We thank the reviewer for this remark and have added a new reference (Ref 36: Younossi ZM, Stepanova M, Zeuzem S, et al. Patient-reported outcomes assessment in chronic hepatitis C treated with sofosbuvir and ribavirin: the VALENCE study. J Hepatol. 2014;61(2):228‐234. doi:10.1016/j.jhep.2014.04.003) (line 366)

11) Corresponding author is not the same in the Title Page and the Submitted form. 

Answer: David Semela is the corresponding author and has submitted the manuscript.

Reviewer #2: 

1) Please provide a better images of figures. It is impossible to read numbers. 

Answer: All figures have been improved and adjusted accordingly. 

2) Authors can give the statistical data about the change in the fatigue parameters and comment on the significance of this reduction. 

Answer: We have added the requested data in the discussion “In this study, treatment with 3D regimen reduced fatigue over time in non-cirrhotic, treatment naïve patients with HCV genotype 1 infection. A reduction in mean FSS score of 2.8 (95% CI 2.21 – 3.43) from baseline (Day 1) to 12 weeks post-treatment (Day 168) was observed in the mITT population. …” followed by: “This change was significant with p < 0.001, Friedman’s ANOVA.” (lines 311-312)

3) It is not true that fatigue was not investigated earlier in this field (line 59 “ To date, there has been no research into the effect of 3D regimen on the quality of life of HCV patients suffering from fatigue.”) because there is already a published paper about this.( Durcan E, Hatemi I, Sonsuz A, Canbakan B, Ozdemir S, Tuncer M. The effect of direct antiviral treatment on the depression, anxiety, fatigue and quality-of-life in chronic hepatitis C patients. Eur J Gastroenterol Hepatol. 2020 Feb; 32(2): 246-250 doi:10.1097/MEG.0000000000001501. PubMed PMID: 31441798.) 

Answer: We thank the reviewer for this comment and have added the publication from Durcan et al.(reference 20) and commented accordingly (lines 66-71): “To date, there has been only few publications on the characterization of fatigue through treatment with newly developed direct-acting antivirals19, and no research into the effect of 3D regimen on the quality of life of HCV patients suffering from fatigue. A recently published work of Durcan et al. found, that direct antivirals did not lead to depression, anxiety or fatigue and did not decrease liver-specific quality of life.”

4) The fatigue is a common symptom in chronic liver disease even if it is not related to viral infection for example primary biliary cholangitis, because of that it is not sense to say “Reduced quality of life in HCV infected patients is independent of liver damage, indicating that the virus itself is responsible.” The virus is not responsible, chronic liver disease is responsible of reduced quality of life. 

Answer: We agree with the reviewer and have adjusted the sentence (lines 58-59): “Reduced quality of life in HCV infected patients is independent of liver damage.”

Reviewer #3: 

An observational single-arm study aimed to measure the impact of 3D regimen treatment on fatigue in Hepatitis C Virus infected patients (n=41). Nearly all patients maintained a virologic response to treatment at 12 weeks, and compared to baseline the fatigue severity scale score decreased at 12 weeks. No changes were observed in daytime physical activity or sleep efficiency. The manuscript was clearly written.

Minor revisions:

1) Line 124: Modify the sentence for clarity. “The primary outcome variable was a change in mean daytime physical activity….” 

Answer: We have adjusted the sentence accordingly (line 145).

2) Line 155: Indicate the statistical method used to calculate the 95% CI. 

Answer: We have clarified this aspect (line 179)

Calculation of 95% CI

The given 95% CIs were calculated as standard Wald intervals using the estimated standard error. In case of sample size n ≤ 40 Brown et al. (2011) recommended the Wilson interval using the null standard error instead of estimated standard error. 

• 93.5 % (CI95%: 89.2 – 97.8) in the ITT (n = 41)

• 92.9 % (CI95%: 88.1 – 97.6) in the scale down ITT (n = 37) 

• 89.5 % (CI95%: 82.5 – 96.5) in the mITT (n = 24). 

References: 

Brown LD, Cai T and DaGupta A (2011). Interval estimation for binominal proportion. Statistical Science. 16 (2): 101–133

Wilson, EB (1927). Probable inference the law of succession and statistical inference. 

J Amer Stat Assoc 22 (158): 209-212

3) Paragraph beginning at line 156: Consider building models to predict daytime physical activity, FSS and sleep efficiency. 

Answer: An analysis of several predictors to the three outcome variables daytime physical activity, sleep efficiency and FSS were performed via generalized linear models with repeated measurement. These new analyses have been added in the supplement (tables S1 and S2, line 270-282):

“An analysis of several predictors to the three outcome variables daytime physical activity, sleep efficiency and FSS were performed via generalized linear models with repeated measurement. As predictors age (classified by median split: ≤ 50 years/>50 years), gender (male/female), HCV genotype (genotype 1a/genotype 1b), liver fibrosis (yes/no) and ribavirin use (yes/no) were investigated, results are shown for sdITT population (supporting information S1 Table) and for mITT population (supporting information S2 Table). Overall, for none of the analyzed possible predictive factors the univariate as well as the multivariate analysis showed significance. The corresponding effect sizes demonstrated no or very small effects for these factors. Therefore, the factors age class, sex, fibrosis, HCV genotype and ribavirin use had no influence of the three outcome variables mean daytime physical activity, sleep efficiency and FSS.”

4) Line 161: Indicate if adverse events were collected according to a standardized method. 

Answer: We have clarified this point (line 186): “Safety assessments were performed in a standardized method at each study visit and included the evaluation of adverse events (AEs):…”

5) Line 192: Intraclass correlation coefficients may be superior to Spearman rank correlation coefficients due to repeated measures. 

Answer: Please find in the following table the correlation coefficients of Fatigue Severity Scale and mean daytime physical activity. 

The authors find the statement in the manuscript sufficient (“There was no correlation between daytime physical activity and FSS score over the course of the study as assessed by Spearman’s rank correlation coefficient.”).

Population Visit n rSpearman* Interpretation

Scale down IT Baseline (V2) 35 0.300 Fair agreement

 Day 28 (V3) 33 0.131 Poor agreement

 Day 84 (V4) 33 -0.022 Poor agreement

 Day 168 (V5) 28 0.195 Poor agreement

Subgroup 1, scale down ITT Baseline (V2) 10 0,468 Moderate agreement

 Day 28 (V3) 10 0,170 Poor agreement

 Day 84 (V4) 9 -0,268 Fair agreement

 Day 168 (V5) 6 -0,493 Moderate agreement

Subgroup 2, scale down ITT Baseline (V2) 13 0,220 Fair agreement

 Day 28 (V3) 13 0,203 Fair agreement

 Day 84 (V4) 12 0,473 Moderate agreement

 Day 168 (V5) 9 0,200 Fair agreement

Subgroup 3, scale down ITT Baseline (V2) 12 0,025 Poor agreement

 Day 28 (V3) 9 0,444 Moderate agreement

 Day 84 (V4) 10 0,176 Fair agreement

 Day 168 (V5) 11 0,555 Moderate agreement

* Spearman correlation coefficient 

Alternatively according to Bland and Altmann (1995) a multiple regression with subjects as a factor was performed to analyze whether the change of mean daytime physical activity in one subject during the study is paralleled by the change of FSS during the study resulting in the following correlation coefficients: sdITT = 0.1477

mITT = 0.1961.

Reference: Bland JM, Altman DG (1995). Calculating correlation coefficients with repeated observations. BMJ. 310: 446

The ICC, addressed by the reviewer, seems to be more appropriate for single variables measured by more than one investigator (e.g. raters) or variables within the same class which is not applicable in the HEMATITE study.

6) Line 221 states, "Concomitant ribavirin, gender, fibrosis stage, age and genotype 1 subtype had no effect on daytime physical activity, FSS score or sleep efficiency in the sdITT or mITT populations." The conclusion in this sentence cannot be supported by results from t-tests or Mann-Whitney-U tests.

Answer: We agree with this comment. An analysis of several predictors to the three outcome variables daytime physical activity, sleep efficiency and FSS were performed via generalized linear models with repeated measurement. These additional results are now described and presented in supplementary tables (S1 and S2), (lines 270-282):

“An analysis of several predictors to the three outcome variables daytime physical activity, sleep efficiency and FSS were performed via generalized linear models with repeated measurement. As predictors age (classified by median split: ≤ 50 years/>50 years), gender (male/female), HCV genotype (genotype 1a/genotype 1b), liver fibrosis (yes/no) and ribavirin use (yes/no) were investigated, results are shown for sdITT population (supporting information S1 Table) and for mITT population (supporting information S2 Table). Overall, for none of the analyzed possible predictive factors the univariate as well as the multivariate analysis showed significance. The corresponding effect sizes demonstrated no or very small effects for these factors. Therefore, the factors age class, sex, fibrosis, HCV genotype and ribavirin use had no influence of the three outcome variables mean daytime physical activity, sleep efficiency and FSS.”

Reviewer #4: 

Thank you very much for your paper. The topic is very important and is also a good investigation. I suggest it might need revision before being accepted. 

1) The study design section is confusing: it is not clear when the study visit 1 took place. 

Answer: Thank you for your comment, which we have clarified: V1 is on day -28. For better comprehension, we have described Visit 1 and Visit 2 as follows in the manuscript: Visit 1 (Day -28; before treatment start), Visit 2 (Day 1; treatment start). Together with the supporting information (Fig S1), we think the study design is now more comprehensive (lines 82-83).

2) The title can be revised as "...... observational study to measure the fatigue....", since the study did not measure the quality of life but measured only fatigue. 

Answer: We prefer to keep the current title, since fatigue is a central element of quality of life. The term “quality of life” is commonly used in the literature assessing fatigue in patients and would in our opinion help to increase the visibility of this study.

3) The small sample size can be mentioned as a limitation of the study.

Answer: We have added this aspect in the discussion (“The relatively small patient number and the inclusion of only F0 and F1 patients further limit our observations.”).

---

## [Decision Letter · Decision Letter 1]

27 Aug 2020

PONE-D-20-05447R1

A prospective, multicenter, post-marketing observational study to measure the quality of life of HCV genotype 1 infected, treatment naïve patients suffering from fatigue and receiving 3D regimen: the HEMATITE study.

PLOS ONE

Dear Dr. Semela,

Thank you for submitting your manuscript to PLOS ONE. After careful consideration, we feel that it has merit but does not fully meet PLOS ONE’s publication criteria as it currently stands. Therefore, we invite you to submit a revised version of the manuscript that addresses the points raised during the review process.

Please, include a comment about the baseline fatigue severity of the patients included in th study.

We look forward to receiving your revised manuscript.

Kind regards,

Jose Ignacio Herrero

Academic Editor

PLOS ONE

Additional Editor Comments (if provided):

The authors have addressed all our previous comments. There is only a detail that should be revised. Please, add a comment in the abstract about the baseline fatigue score (severe in all the patients) included in the study.

Reviewers' comments:

Reviewer's Responses to Questions

**Comments to the Author**

1. If the authors have adequately addressed your comments raised in a previous round of review and you feel that this manuscript is now acceptable for publication, you may indicate that here to bypass the “Comments to the Author” section, enter your conflict of interest statement in the “Confidential to Editor” section, and submit your "Accept" recommendation.

Reviewer #2: All comments have been addressed

Reviewer #3: All comments have been addressed

Reviewer #4: All comments have been addressed

2. Is the manuscript technically sound, and do the data support the conclusions?

Reviewer #2: Yes

Reviewer #3: (No Response)

Reviewer #4: Yes

3. Has the statistical analysis been performed appropriately and rigorously? 

Reviewer #2: Yes

Reviewer #3: (No Response)

Reviewer #4: Yes

4. Have the authors made all data underlying the findings in their manuscript fully available?

Reviewer #2: Yes

Reviewer #3: (No Response)

Reviewer #4: Yes

5. Is the manuscript presented in an intelligible fashion and written in standard English?

Reviewer #2: Yes

Reviewer #3: (No Response)

Reviewer #4: Yes

6. Review Comments to the Author

Reviewer #2: (No Response)

Reviewer #3: (No Response)

Reviewer #4: The authors addressed all the questions. It is written in standard English. The authors made all the underlying findings in their manuscript fully available.

7. PLOS authors have the option to publish the peer review history of their article (what does this mean?). If published, this will include your full peer review and any attached files.

Reviewer #2: No

Reviewer #3: No

Reviewer #4: **Yes: **Tatevik Balayan

---

## [Author Response · Author response to Decision Letter 1]

7 Oct 2020

Dear Reviewers,

Thank you for reviewing our revised manuscript ‘A prospective, multicenter, post-marketing observational study to measure the quality of life of HCV genotype 1 infected, treatment naïve patients suffering from fatigue and receiving 3D regimen: the HEMATITE study.’

There has been raised one point and we did amend the abstract (in Methods and Results) accordingly, traceable in the track changed version of the manuscript:

Question by the editor: The authors have addressed all our previous comments. There is only a detail that should be revised. Please, add a comment in the abstract about the baseline fatigue score (severe in all the patients) included in the study.

Answer: We thank you for this valuable comment. Accordingly, we have added the information “Patients with FSS ≥ 4 (clinically significant fatigue) were included.” in lines 23-24 and “Mean baseline FSS score was 6.0 for the sdITT population and 5.9 for the mITT population and decreased from baseline to 12 weeks post-treatment by 2.6 (95% confidence interval [CI]: 2.1, 3.1) for the sdITT (n=37) population and 2.8 (95% CI: 2.2, 3.4) for the mITT (n=24) population.” in lines 30-33 of the abstract. 

We hope that all your questions have been addressed adequately. We strongly believe that this study will be of significant interest for the readership of PLOS ONE. Thank you once again for considering this manuscript for publication in PLOS ONE. 

Best regards,

David Semela, on behalf of the authors

---

## [Editor Report · Decision Letter 2]

13 Oct 2020

A prospective, multicenter, post-marketing observational study to measure the quality of life of HCV genotype 1 infected, treatment naïve patients suffering from fatigue and receiving 3D regimen: the HEMATITE study.

PONE-D-20-05447R2

Dear Dr. Semela,

We’re pleased to inform you that your manuscript has been judged scientifically suitable for publication and will be formally accepted for publication once it meets all outstanding technical requirements.

Kind regards,

Jose Ignacio Herrero

Academic Editor

PLOS ONE

Additional Editor Comments (optional):

The authors have given adequate responses to all the previous comments
---

## [Editor Report · Acceptance letter]

23 Oct 2020

PONE-D-20-05447R2 

A prospective, multicenter, post-marketing observational study to measure the quality of life of HCV genotype 1 infected, treatment naïve patients suffering from fatigue and receiving 3D regimen: the HEMATITE study. 

Dear Dr. Semela:

I'm pleased to inform you that your manuscript has been deemed suitable for publication in PLOS ONE. Congratulations! Your manuscript is now with our production department. 

Kind regards, 

on behalf of

Dr Jose Ignacio Herrero 

Academic Editor

PLOS ONE